# Psychiatric Neural Networks and Precision Therapeutics by Machine Learning

**DOI:** 10.3390/biomedicines9040403

**Published:** 2021-04-08

**Authors:** Hidetoshi Komatsu, Emi Watanabe, Mamoru Fukuchi

**Affiliations:** 1Medical Affairs, Kyowa Pharmaceutical Industry Co., Ltd., Osaka 530-0005, Japan; 2Department of Biological Science, Graduate School of Science, Nagoya University, Nagoya City 464-8602, Japan; 3Interactive Group, Accenture Japan Ltd., Tokyo 108-0073, Japan; emi-w@ruri.waseda.jp; 4Laboratory of Molecular Neuroscience, Faculty of Pharmacy, Takasaki University of Health and Welfare, Gunma 370-0033, Japan; fukuchi@takasaki-u.ac.jp

**Keywords:** psychiatric disorder, machine learning, neural network, antipsychotics, schizophrenia, bipolar disorder, depression, precision medicine, endophenotype, decision making

## Abstract

Learning and environmental adaptation increase the likelihood of survival and improve the quality of life. However, it is often difficult to judge optimal behaviors in real life due to highly complex social dynamics and environment. Consequentially, many different brain regions and neuronal circuits are involved in decision-making. Many neurobiological studies on decision-making show that behaviors are chosen through coordination among multiple neural network systems, each implementing a distinct set of computational algorithms. Although these processes are commonly abnormal in neurological and psychiatric disorders, the underlying causes remain incompletely elucidated. Machine learning approaches with multidimensional data sets have the potential to not only pathologically redefine mental illnesses but also better improve therapeutic outcomes than DSM/ICD diagnoses. Furthermore, measurable endophenotypes could allow for early disease detection, prognosis, and optimal treatment regime for individuals. In this review, decision-making in real life and psychiatric disorders and the applications of machine learning in brain imaging studies on psychiatric disorders are summarized, and considerations for the future clinical translation are outlined. This review also aims to introduce clinicians, scientists, and engineers to the opportunities and challenges in bringing artificial intelligence into psychiatric practice.

## 1. Introduction

Living organisms have self-sustaining properties that are absent in purely physical systems. They acquire energy from food for sustenance, growth, and reproduction. Another self-sustaining ability of animals is the adaptive behavior. Behavioral strategies that improve their ability to acquire energy and produce successful offspring to help each species proliferating evolutionarily through the process of natural selection [1]. Three essential principles for biological behaviors have been proposed as materiality (an embodied brain embedded in the world), agency (action-perception closed loops and purpose), and historicity (individuality and historical contingencies) [2]. The three principles are arguably unique to life and imperative for neuroscience. We believe that these considerations will shed light on our typical approaches to address not only the conundrum of animal behaviors but also the nature of mental disorders.

The complexity of environment and social dynamics often make it difficult to identify optimal behaviors in the real world. Decision-making in this process involves many different brain areas and circuits that are exacerbated in numerous neurological and psychiatric disorders [3]. Traditional approaches have dominated the studies on optimal behaviors. Among these, a prescriptive approach addresses the question of what is the best choice for a given challenge. For instance, economists and game theorists describe how self-interested rationales should behave individually or in a group [4,5]. However, the real behaviors of animals and humans seldom follow the predictions of such prescriptive theories. Indeed, prospect theory can predict the decision-making of animals as well as humans more accurately than the prescriptive theories [6,7,8]. Likewise, humans often behave altruistically and thus deviate from the predictions of the game theory [9,10]. Recently, these traditional approaches have merged with neuroscientific theories, in which learning plays a crucial role in choosing optimal behaviors and decisions. Particularly, reinforcement learning theory presents a worthwhile framework to model how an individual’s behaviors are tuned by experience [11,12]. Neuroscientists have begun to uncover numerous core mechanisms in the brain responsible for various computational processes of learning and decision-making. Their findings are now frequently featured in the literatures in many disciplines, such as ethics [13], law [14], politics [15], marketing [16], and economic and financial decisions [17,18].

A distinct set of computational algorithms evoked through coordination among myriad brain systems are abnormal in many types of neurological and psychiatric disorders, leading to aberrant and maladaptive behaviors [19,20,21,22,23,24]. Notwithstanding, psychiatric status is still diagnosed and treated according to the experiential schemes based on symptomatic phenotypes. The definitions of many mental disorders described in the DSM and ICD manuals do not always match well with neuroscientific, psychopathological, and genetic evidences [25,26]. Thus, there is a greater desire to redefine mental illness as a discrete disease. To satisfy this aspect, the Research Domain Criteria (RDoC) initiative has been launched to reconceptualize mental disorders as a dimensional approach that incorporates many different levels of data from molecular factors to social determinants and linked more precisely to interventions for a given individual [27,28,29]. This approach is more likely to be compatible with the facts that psychiatric patients comprise of clusters of psychopathological symptoms and that many symptoms are shared among different mental disorders. Machine learning approaches are well suited to achieve this goal.

Machine learning approaches for psychiatry feature statistical learning functions from multidimensional data sets to unveil general principles underlying a series of observations without definite guidance. Machine learning algorithms can be generally classified into three categories, namely supervised, unsupervised, and reinforcement learning methods [30,31]. Supervised learning models, such as support vector machines (SVM) and neural-network algorithms, are designed to predict a discrete outcome (e.g., healthy group vs. psychiatric group), or continuous outcome (e.g., psychiatric severity degrees) from the qualitative data on behaviors (e. g., questionnaire), genetics (e. g., single nucleotide polymorphisms, gene expressions), or brain function (e. g., neural activity). Unsupervised learning describes models to discover unknown statistical configurations across subjects without reference to a specific outcome. Reinforcement learning investigates how actions in one’s environment (such as treatment) change behaviors [31]. Among these algorithms, supervised learning, especially SVM, is most widely used in psychiatry to classify individuals into groups within a statistical framework. This approach has already shown promising results in neuroimaging-based psychosis prediction and treatment-response estimation [32,33,34]. With increasing digitized phenotypic data, improved computing power, and less expensive data storage, machine learning, as well as deep learning that is the artificial neural network algorithms to learn complex representations of high-dimensional data patterns such as images and language, would offer findings with important implications for the development of therapeutic interventions, leading to precision psychiatry and stratified clinical trial designs.

A main purpose of this review is to exemplify the new insights provided by recent applications of machine learning in neuroimaging and clinical studies on major psychiatric disorders. To this end, decision-making in real life and mental illness is briefly described. Next, our current knowledge of neuronal circuits or functional connectivity in major psychiatric disorders is summarized. This review also argues that machine learning is predisposed to address many challenges in the era of precision psychiatry.

## 2. Decision-Making in Real Life and Psychiatric Disorders

The understanding of decision-making processes helps us develop machine learning tools for precision psychiatry. Flawed decision-making has been commonly observed in major psychiatric disorders, often causing poor real-life outcomes. In many cases, flawed decision-making in mental illness results from abnormalities in fundamental neuropsychological processes, including impaired attention, reward processing, associative learning, and working memory. For instance, defects in reward and avoidance learning are reported in patients with depression. Aberrant hedonic capacity and cognitive impairment occur in bipolar disorder and schizophrenia. Flawed decision-making can contribute to abnormal behaviors such as nonadherence to medications or outpatient appointments, failing to exercise, or poor diet. Downstream consequences of poor decisions can lead to worsening of symptoms, reduced life satisfaction, impaired everyday functioning, relapse and rehospitalization, poor physical health, and even more tragic outcomes such as accidental death, homicide, or suicide [35].

Decision-making is a constant process in real life and takes place from when we wake up until we go to bed. Its processes are largely divided into three steps: (1) identification and depiction of all alternatives, (2) assessment of the consequences of each alternative, and (3) a comparison of the accuracy and efficiency of each of these consequences [35]. Different disciplines attempt to systematize the understanding of decision-making. For instance, consumer decision-making has long been of interest to economists. Consumers are viewed as rational decision-makers who are only concerned with self-interest. One of the most prevalent consumer decision models is the Engel-Blackwell-Miniard Model (Figure 1) [36], in which every conceptual step of decision-making is instrumental for developing machine learning algorithms to implement customized advertising tactics [37]. The model has the following decision processes: need recognition followed by a search of information, the evaluation of alternatives, purchase, and finally, post purchase reflection. These decisions can be influenced by two main factors, namely memories of previous experiences and external variables in the form of either environmental influences or individual differences. In other words, this model shows that various computational steps of decision-making can be affected by the environment, individual factors, and memory. Similarly, the environment and individual factors, including polygenic architecture and epigenetic risk elements, underlie the manifestation of psychiatric disorders [38]. The pathophysiology of psychiatric disorders is complex and not well understood. Thus, it will be interesting to illustrate mental illness with common denominators across diagnostic boundaries. One common element in severe mental illness is pervasive poor decision-making. Since innumerable combinations of computational algorithms in the brain are evoked in a flexible manner for optimal decision-making, it would be challenging to elucidate the nature of decision-making impairments in different psychiatric disorders. Therefore, econometric models are becoming valuable tools for computational psychiatry [26,39,40,41,42].

### 2.1. Schizophrenia

Schizophrenia is a severe psychiatric disorder that affects approximately 1% of the population worldwide. It is characterized by positive symptoms (e.g., delusion, hallucination, and thought disorder), negative symptoms (e.g., apathy, poor social functioning, and emotional blunting), cognitive deficits and other psychopathological symptoms (e.g., psychomotor retardation, lack of insight, poor attention, and impulse control) [43]. These symptoms are likely linked not only to excessive dopaminergic transmissions in the mesolimbic pathway but also to the dopamine release decline in the prefrontal cortex. Although dysfunctions of N-methyl-D-aspartic acid (NMDA) receptor systems and weaker prefrontal GABAergic actions are also implicated, the precise manner in which plural neurotransmitter systems interact with one another in schizophrenia remains elusive [44].

Various types of cognitive functions are impaired in schizophrenia [45]. Altered prefrontal functions might also be responsible for abnormalities in decision-making and reinforcement learning observed in patients with schizophrenia [46]. For instance, during economic decision-making challenges, schizophrenic patients incline to put less weight on potential losses compared to the healthy subjects [47], and also show impairments in feedback-based learning [48,49]. Consistent with these results, activity involved in reward prediction error in striatum and the frontal cortex is attenuated in patients with schizophrenia [50,51]. A pervasive clinical burden in schizophrenia is the high prevalence of comorbidity with substance abuse disorders. Approximately half of patients with schizophrenia exhibit a lifetime history of substance abuse disorders [52], and almost all of them are smokers [53,54]. These unusual high rates of substance-use comorbidity may be attributed to disrupted reward processing [55]. These global cognitive impairments are translated in the devastating functional toll of this disorder.

### 2.2. Bipolar Disorder

Bipolar disorder and schizophrenia share high levels of polygenic and pleiotropic molecular architecture [38]. In accord with this, these two disorders share similar types of impairments in cognitive domains including processing speed, attention, working memory, and executive function, although bipolar disorder usually exhibits less severe deficits [56]. These cognitive deficits bring about a substantial clinical burden in up to 60% of patients with bipolar disorder and can be observed not only in depressed, manic, and mixed episodes but also in the euthymic state. These pervasive impairment in bipolar disorder indicates that it may be a trait marker linked with genetic vulnerability [57]. In addition to cognitive dysfunctions, emotion processing is severely altered in patients with bipolar disorder. Upregulated processing of positive emotion regardless of the context is central to the manic bipolar episode [58]. Theory of mind and emotion processing are significantly disrupted in the euthymic bipolar state [59]. Thus, bipolar patients display defects in their ability to understand other emotions and intentions, with a resultant impact on everyday functioning. Moreover, euthymic bipolar patients show moderate to severe impairments in a broad range of executive functions including mental manipulation, verbal learning, abstraction, sustained attention, and response inhibition [60,61,62,63,64]. In addition to these deficits, there seems to be specific decision-making biases in bipolar patients including impulsivity and deficits in risk assessment and reward processing. The weights of these aberrant decision-making are evident in depressed bipolar patients [35].

### 2.3. Depression and Anxiety Disorders

Depression and anxiety disorder are characterized by disturbances in mood and emotion and feature poor concentration and negative mood states, such as sadness and anger, with high levels of their comorbidities [65,66,67]. However, they have some important differences. Anxiety is needed to improve individual’s readiness for impending danger, whereas depression might prohibit previously unsuccessful actions and enhance more reflective cognitive processes. Both depression and anxiety disorder often cause systematic biases in decision-making [67,68]. Patients with anxiety disorders are hypersensitive to threatening cues without apparent memory bias. In contrast, depressed patients show a bias to memorize negative events and ruminate excessively [69].

The symptoms of these two mood disorders have been extensively investigated, and some responsible brain regions have been identified. For instance, symptoms of depression is associated with abnormalities in frontostriatal monoamines involved in reinforcement learning [70]. Meanwhile, the brain regions responsible for anxiety disorders include the amygdala, insula, and anterior cingulate cortex [71,72]. Interestingly, the default network is overactive in patients with depression. The levels of excessive rumination and negative self-referential memory in depressive states are likely to be correlated with the default network function [73]. Indeed, deep brain stimulation in the subgenual cingulate cortex of patients with major depressive disorder produces therapeutic effects [74].

Although the neurochemical mechanism of mood disorders remains unclear, much attention has been paid to the role of changed serotonin metabolism [75]. For instance, it has been hypothesized that future reward is discounted markedly in patients with depression due to a low level of serotonin [76]. Serotonin has been proposed to be primarily involved in the inhibition of thoughts and behaviors associated with aversive outcomes, including the heuristic process of unpromising decision-making [77,78,79].

### 2.4. Autism Spectrum Disorder

Autism spectrum disorder (ASD) is a neurodevelopmental disorder, featuring early-onset impairments in social cognition, poor communication abilities, restricted repetitive and stereotyped behaviors, and narrow interests that hurt the individual’s ability to function properly in school, work, and other fields of life [80]. Particularly, patients with ASD show impairments in their ability to make inferences about the intentions and beliefs of others, namely, theory of mind [81,82]. Such impaired abilities might underlie differences in socially interactive decision-making tasks between autistic patients and healthy subjects. Autism is known as a “spectrum” disorder because there is a wide range of variations in the type and severity of symptoms [83]. Under the DSM-5 criteria, patients with ASD must display symptoms from early childhood, even if those symptoms are not recognized later. They also may not be fully recognized until social demands surpass their capacity to receive the diagnosis. The earliest symptoms include the lack of attention to faces [84], imitative behaviors [85], and motor deficits [86].

In ASD, a variety of brain architectures are altered, ranging from the brain stem to the cerebellum and cerebral cortex [87,88,89,90]. In particular, the connectivity deficit in the parieto-frontal circuit involved in the mirror mechanism has been proposed to underlie some cognitive aspects of ASD [91,92]. Several brain regions affected in ASD have also been shown to be involved in decision-making. Neurobiological studies have revealed that ASD has functional abnormalities in the amygdala, prefrontal cortex, superior temporal sulcus, and fusiform gyrus whose brain regions are thought to constitute the “social brain” [93,94]. Especially, the amygdala, ventral striatum, and prefrontal cortex are implicated in decision-making according to functional neuroimaging and lesion analyses [95,96,97].

## 3. Psychiatric Neural Networks

Brain imaging can give important insights into the underlying neural mechanisms of psychiatric disorders. For instance, the resting state functional magnetic resonance imaging (rs-fMRI) analyses in humans have identified a large number of potentially important abnormal functional connections that may underlie psychiatric manifestations [98,99,100,101]. Most of MRI-based neuroimaging biomarker studies have applied the machine learning algorithm of SVM to achieve high accuracy with many features and shown that different diagnoses are related to unique patterns of functional connections [102,103,104,105].

Functional connectivity (FC) shows how brain regions are temporally coordinated and is becoming more and more used to investigate neuronal network architecture. Resting state FC has been linked with a diverse range of individual traits [106,107,108,109,110]. For instance, whole-brain FC models have revealed that patterns of functional connections across brain regions can predict cognitive abilities not only in healthy individuals but also in individuals with mental illness [111,112,113,114,115]. Intriguingly, a prediction model of working memory on letter three-back task performance using whole-brain FC shows the order of working memory impairment for major psychiatric disorders (i.e., schizophrenia > major depressive disorder > obsessive-compulsive disorder > ASD) [115]. This suggests that specific cognitive processes may be represented by the corresponding FC patterns among distributed neuronal networks. Whole-brain FC-based models also have been shown to predict psychiatric disease, including schizophrenia, ASD, and major depressive disorder, as well as individual clinical severities [105,116,117], suggesting that FC alteration is quantitatively associated with psychiatric abnormality.

MRI-based delineation of psychiatric disorders has been explored as a complement to the current symptom-based diagnoses. While a number of studies have identified numerous disease-specific functional and structural aberrations, none of them are practically used as a credible biomarker mostly because of the lack of its generalizability. Namely, the reliability of the developed classifiers has not been demonstrated with regard to the variety of population demographics and data attributes [118,119,120,121,122,123,124]. These elements include different ethnicities, sex, ages, medication profiles, scanner specifications, imaging parameters, and instructions to participants, all of which are known to affect the MRI results [108,125,126,127,128,129]. Thus, little attention has been paid to the neuroimaging-based biomarkers in neuropsychiatry until recently [130,131].

To identify a generalizable classifier, we must surmount the following two major difficulties: over-fitting and nuisance variables (NVs). First, certain situations in data and model properties bring about over-fitting problems where the model fitting to the training data can be so precise that the associated errors become artificially smaller compared with the inherent data variance [130]. Second, any machine learning algorithms used for classification is doomed to employ NVs specific to a given sample data and to falsely choose neuroimaging features that are associated with the NVs. NVs include site-specific conditions in image acquisition and properties in the sample population such as demographic attributes, treatment status, and illness severity. To abrogate the over-fitting and the effects of NVs, advanced approaches with a unique combination of machine learning algorithms across multiple imaging sites have recently identified generalizable FC classifiers correlated with specific psychiatric disorders as described below.

### 3.1. Functional Connectivity as ASD Classifier

The rs-fMRI studies have revealed a reliable neuroimaging-based classifier for ASD that shows the spatial distribution of the 16 FCs identified from the data at multiple sites in Japan by the machine-learning algorithm (Figure 2 and Table 1) [105]. They also have demonstrated that both a sophisticated machine learning algorithm and a large training data set are prerequisite for identifying a reliable and generalizable classifier. Concerning the hemispheric distribution of the 16 FC related-brain regions, there are significantly more regions in the right hemisphere than in the left. Concerning the functional network attributes of the 32 brain regions constituting these 16 FCs, the 13 brain regions participate in the cingulo-opercular network [105,108,132]. This ASD classifier achieves a diagnosis prediction accuracy of 85% for each individual with balanced sensitivity and specificity of 80% and 89%, respectively [105]. Intriguingly, the ASD classifier is intermediately generalizable to schizophrenia, whereas it hardly exhibits any generalizability to attention-deficit hyperactivity disorder (ADHD) and major depressive disorder (MDD). This suggests that ASD shares more intrinsic neural networks with schizophrenia than with ADHD or MDD [105]. In concordance with this, genome-level studies have found that ASD shares a high degree of polygenic risk with schizophrenia, but not with ADHD or MDD [133,134]. Accumulating evidence by clinical and behavioral studies also has shown a close relationship between ASD and schizophrenia [135,136].

### 3.2. Functional Connectivity as Schizophrenia Spectrum Disorder Classifier

The rs-fMRI studies also have identified a reliable classifier for schizophrenia spectrum disorder (SSD) using L1-norm regularized sparse canonical correlation analyses and sparse logistic regression (SLR) [137,138]. The machine-learning algorithms automatically selected SSD-specific FCs from about 10,000 FCs of whole-brain rs-fMRI [116]. The SSD classifier shows the distinctive 16 FCs that are distributed as interhemispheric (44%), left intra-hemispheric (25%), and right intra-hemispheric connections (31%) (Figure 2 and Table 1) [116]. The classifier differentiates SSD from healthy controls with an accuracy of 76% [116]. The 16 FCs as SSD classifier are different from the 16 FCs as ASD classifier mentioned above (Figure 2d). The weighted linear summation (WLS) of the selected FCs predicts the categorical diagnostic label for each individual. The values of WLS provide a degree of classification certainty, which can be construed as neural classification certainty for the disorders. Then, each biological dimension can be determined based on the WLS [116]. On the basis of the SSD and ASD biological dimensions, the WLS distributions of individuals with SSD and ASD display overlapping but asymmetrical patterns in the two biological dimensions. This suggests that the neuronal network of SSD is characterized by a larger diversity and that it partially shares spatial distributions with the smaller network of ASD. In accord with this, the recent genetic findings demonstrate that ASD shares a significant degree of polygenic architecture with SSD [133], and that common genetic variants explain nearly 50% of total liability to ASD and approximately 30% of total liability to SSD [139,140].

### 3.3. Functional Connectivity as MDD Classifier

According to the meta-analysis published results of MRI-based biomarkers in depressive disorders [141], approximately 30% of them harnessed rs-fMRI as modality, of which only one-third employed FCs among region of interests (ROIs). As most of those studies have applied the SVM algorithm to achieve high diagnostic accuracy, it remains unknown that which are the most critical FCs in depression across the whole brain. Approximately half of depressed patients are inadequately treated by available interventions, as there are no reliable guidelines to match patients to optimal treatments. This mainly derives from the heterogeneity of depression [142]. Thus, it would be important to pay attention to a specific subtype of depression in order to identify target FCs.

Melancholic major depressive disorder is a subtype of MDD that is considered to be the most drug-responsive [143,144,145,146]. The sparse machine learning algorithm identified melancholic MDD-specific 10 FCs from rs-fMRI data of 130 individuals including melancholic MDD patients and healthy controls (Figure 2 and Table 1) [117]. Importantly, this biomarker does not generalize to non-melancholic and treatment-resistant MDD, ASD, and schizophrenia. Out of 10 FCs, the top two FCs as the melancholic MDD-specific classifier includes the FCs (SN-ECN connectivity) with left inferior frontal gyrus (IFG) in executive control network (ECN) and right dorsomedial prefrontal cortex (DMPFC)/frontal eye field (FEF)/supplementary motor area (SMA) in salience network (SN), and the FCs (DMN-ECN connectivity) with left dorsolateral prefrontal cortex (DLPFC)/inferior frontal gyrus (IFG) in executive control network (ECN) and posterior cingulate cortex (PCC)/Precuneus in default mode network (DMN). These brain regions are tightly linked with cognitive flexibility, such as reversal learning tasks, in which patients with depression often have functional impairments [147,148]. In SN-ECN connectivity, activation in IFG and DMPFC leads to empathic accuracy with compassion meditation training [149], and adolescents with depression show reduced connectivity between DMPFC and IFG during cognitive reappraisal of emotional images [150]. Priming transcranial magnetic stimulation (TMS) studies demonstrate that the DMPFC play an essential role in forming social-relevant impression, such as processing verbal emotional stimuli and face-adjective pair [151,152]. In DMN-ECN connectivity, bilateral DLPFC/IFG relate to conflict processing and attention control [153], and PCC/Precuneus are linked with anhedonic anxious arousal and depression [154]. In addition to the accumulated evidence that links the connections of DLPFC to depression [110,155,156,157,158,159,160,161], there is another evidence that supports therapeutic relevance. For instance, DLPFC is a well-known target for repetitive TMS (rTMS) therapy for treatment-resistant depression [162,163,164]. Besides, neurofeedback therapy targeting DMN-ECN connectivity has been reported to be effective [165,166,167], suggesting that DLPFC plays a causal role in manifestation of depression.

## 4. Machine Learning Approach to Predict Therapeutic Outcomes in Psychiatric Disorders

Psychiatric research and treatment are based on a diagnostic system exclusively dependent on human experiential terms rather than on objective biological markers. Psychiatrists use a prolonged trial-and-error process to identify the optimal medications for each patient [168,169]. Although the standard diagnostic classifications have been constructed from expert opinions and defined in DSM and ICD, they are not sufficient for judging an appropriate treatment for each patient. Modern drug treatment choices are only effective in roughly half of the patients [25,141]. This is because of the heterogeneity of psychiatric disorders and the unknown precise mechanism of action of antipsychotic drugs. The psychiatrist’s choice from the best-possible treatment options does not rely on what has caused or maintained the mental illness of a given patient. While current clinical research goal is mainly to discover novel treatment options that benefit some majority of a certain patient group, an attractive alternative research goal is to improve the choice from existing treatment options by predicting their effectiveness of individual patients. In fact, a specific drug or psychotherapy treatment has been successful in a particular clinical group and unsuccessful in another patient group labeled even with the identical diagnosis [170]. This approach might help reduce the time spent in trial-and error treatment and accompanying personal and economic burden.

Machine learning methods can offer a set of tools that are especially suited to achieve clinical predictions at the individual level. Predictive models are conceptually positioned between genetic vulnerability as an individual’s architecture and clinical symptoms as an individual’s behavioral manifestations. The exploitation of various endophenotypes has the translational potential to refine individual clinical management by early diagnosis, disease stratification, optimal choices of drug treatments and dosages, and prognosis for psychiatric care (Figure 3) [99]. Machine learning models have a long-standing focus on prediction as a metric of statistical quality and are able to predict an outcome from single observation, such as behavioral, neural, or genetic measurements of individuals [141,171]. In contrast, traditional statistical methods, such as Student’s t test, are often used in medical research to explain variance of group effects.

An observed impact evaluated to be statistically significant by a *p* value does not always produce a high prediction accuracy in new and independent data. A classical null-hypothesis method takes a one-step procedure. Namely, a given dataset is routinely used to yield a *p* value or an effect size in a single process. This result itself cannot be used to judge other data in later steps. In a two-step procedure of machine learning models, a learning algorithm is fitted on a larger amount of available data (training data) and the resulting “trained” learning model is evaluated by application to new data (test data). In a first step, structured knowledge in openly available or hospital-provided data sets can be extracted. In a second step, the resulting trained algorithms can be applied with little effort in a large number of individuals in diverse mental health contexts.

### 4.1. Prediction of Therapeutic Outcomes in Schizophrenia

Despite the established pharmacological treatments for schizophrenia, up to 50% of the patients develop poor disease outcomes [172,173]. Stratifying treatment through the early recognition of outcome indicators might alleviate unfavorable disease progression in these patients [174,175]. Group-level studies have discovered many potential outcome predictors, such as disease course variables, treatment adherence and response, comorbidity, and cognitive impairments [176]. It remains unclear which of these characteristics should be combined for prediction, whether these group-level findings can be used to generate significant predictions for individual patients, or how accurate these predictions might be at another sites. Furthermore, the outcome can be not merely defined by symptomatic remission, but has to include various concepts that focus on restored functioning to widely cover treatment effects [177].

Large multisite treatment databases containing prospective phenotypic data of psychiatric disorders, such as the European First Episode Schizophrenia Trial (EUFEST), enable us to develop powerful machine learning methods to reliably predict treatment outcomes. By using data from EUFEST, the pooled non-linear SVMs predicted patients’ 4-week and 52-week outcomes with cross-validated balanced accuracies (BAC) of 75.0% and 73.8%, respectively [34]. This non-linear SVMs surpassed linear SVMs, univariate and multivariate logistic regression, and decision tree ensembles. Intriguingly, the most useful predictors of poor 4-week and 52-week treatment outcome were unemployment, daytime activities, psychological distress, educational difficulties, low educational status of the patient and patient’s mother. In patients with good 52-week prognosis, haloperidol was linked with shorter adherence compared with ziprasidone, amisulpride, and olanzapine due to insufficient response and side effects. Olanzapine and amisulpride were associated with higher global assessment of functioning (GAF) scores than quetiapine [34].

Recent machine learning methods have revealed that functional striatal abnormalities (FSA) are significantly correlated with a spectrum of severity across psychiatric disorders, where dysfunction is most severe in schizophrenia, milder in bipolar disorder, and indistinguishable from individuals in obsessive-compulsive disorder (OCD), depression, and ADHD [178]. FSA scores provide a personalized index of striatal dysfunction and distinguish individuals with schizophrenia from healthy controls with an accuracy exceeding 80% (sensitivity, 79.3%; specificity, 81.5%). FSA scores are also significantly linked with antipsychotic treatment response. Interestingly, clozapine, the only drug approved for treatment-resistant schizophrenia, is not associated with FSA scores. This may suggest that FSA can characterize treatment response across different types of antipsychotics, preferentially the types with lower Meltzer’s ratio (5-HT2A/D2 affinity ratio) [178].

### 4.2. Prediction of Therapeutic Outcomes in Depression

Antidepressant medication is the first option for the treatment of MDD. However, remission rates are approximately 30% to 55% after first or second medication trials and then drop with subsequent medication trials [168]. Thus, the patients experience trial and error periods with different treatments before finding the optimal one. One solution is to identify the biological predictors of response to an antidepressant. This may expedite the treatment and lead to faster relief of the symptoms.

Electroencephalography (EEG) and fMRI have been used for predicting treatment response in MDD patients [179,180,181]. Since EEG is relatively more available and cost-effective than fMRI, it is a good option for developing such treatment biomarkers. The accumulated findings suggest that EEG-derived features before treatment may predict clinical response to antidepressants [182,183]. Several EEG studies have shown that characteristics of resting-state neural oscillations, especially in alpha and theta bands, may predict the drug response [184,185,186,187,188]. For instance, posterior alpha activity is associated with response to amitriptyline and fluoxetine [189,190]; theta activity with imipramine, venlafaxine, and several selective serotonin reuptake inhibitors (SSRI) [188,191,192]; interhemispheric delta asymmetry with fluoxetine [193]; delta activity with paroxetine and imipramine [188,194]; delta activity in rostral anterior cingulate cortex with nortriptyline, venlafaxine, and fluoxetine [195,196]; nonlinear features of EEG signals with clomipramine, escitalopram, citalopram, bupropion, and mirtazapine [197,198,199].

Although resting-state EEG can be useful for predicting drug response, the aforementioned studies fail to address several questions that are important for translating this finding into a clinical tool. First, they use small and homogeneous sample sets. Second, most of them report low prediction accuracy. Third, their prediction accuracies are biased upward by the lack of an independent testing set. One of the most powerful methods for addressing these questions is applying machine learning techniques. Zhdanov et al. used a SVM classifier to predict EEG-derived escitalopram treatment outcome using data from a large, Canada-wide, multicenter study, the first Canadian Biomarker Integration Network in Depression (CAN-BIND-1) study [200]. This classifier identified responders with an estimated balanced accuracy of 79.2% (sensitivity, 67.3%; specificity, 91.0%) when using EEG data recorded before the treatment, whereas additional EEG data after first 2 weeks of treatment increased the accuracy to 82.4% (sensitivity, 79.2%; specificity, 85.5%). In rTMS, Hasanzadeh et al. used k-nearest neighbors (KNN) classifier to predict its EEG-derived treatment response in MDD [201]. Using beta bands, this classifier discriminated between responders and non-responders to rTMS treatment with the accuracy of 91.3% (sensitivity, 91.3%; specificity, 91.3%) when resting-state EEG data from 46 MDD patients were used.

## 5. Conclusions

The infusion of economic and machine learning framework into neuroscience has rapidly advanced our understanding of neural mechanisms for various cognitive processes including decision-making. Because flawed decision-making is the most prominent symptoms in numerous psychiatric disorders, it is essential for neuroscientists and psychiatrists to combine their expertise to develop more effective treatment. They also need to redefine mental illness to meet biological and pathological evidence as seen in the RDoC initiative, while DSM/ICD manuals often reflect public values, such as the definition of sexual identity disorders and the advent of internet gaming disorder [202,203], and have been frequently revised so far. Since substantial progress in major mental illness research has been made in understanding the molecular mechanisms through basic and translational approaches, including cell and animal models, those types of big data may also be helpful for reinforcing weakness of RDoC [204]. Following the growing data richness and changing research questions, machine learning or deep learning algorithms could enable clinical translation of empirically trusted prediction for individual patients in a fast, cost-effective, and practical manner. Such artificial intelligence algorithms may be particularly tuned to precision psychiatry because they can directly translate large-scale multidimensional data into clinical relevance. From a long-term and larger perspective, it is particularly challenging to verbalize mechanistic hypotheses for mental disorders at the abstraction level, ranging from molecular mechanisms to urbanization trends in society. Individual consumer decision patterns would be useful for this purpose. Ultimately, we may more effectively impact psychiatric disorders that arise from the interplay between genetic vulnerability and life experience, both of which are unique to each individual.

## Figures and Tables

**Figure 1 biomedicines-09-00403-f001:**
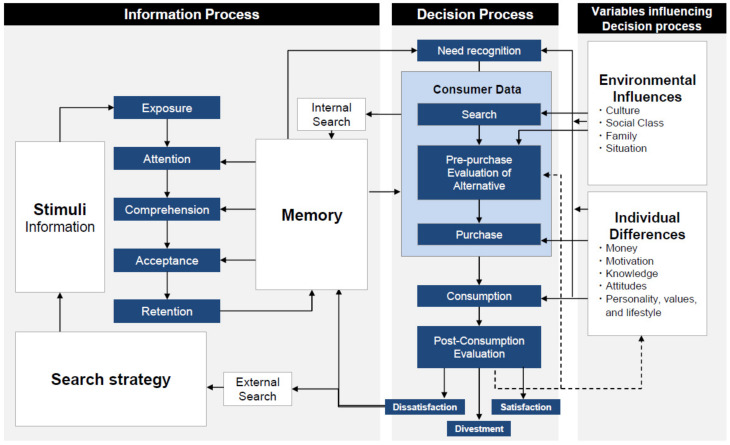
Schematic illustration of the Consumer Decision Model. The Consumer Decision Model (also known as the Engel-Blackwell-Miniard Model) that was originally developed by Engel, Kollat, and Blackwell is modified [36]. This model comprises conceptual steps of decision making in the real world and thus could be extended to be applied for developing precision medicine in psychiatry.

**Figure 2 biomedicines-09-00403-f002:**
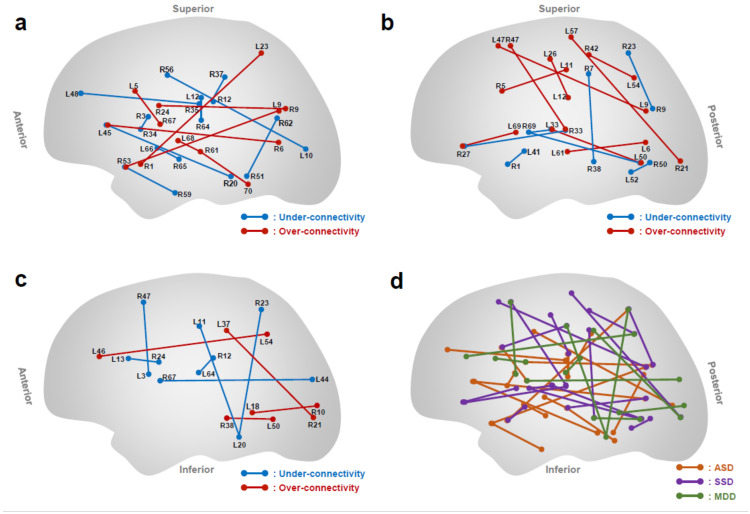
The specific FCs identified for ASD/SSD/MDD classifiers. The 70 terminal regions connected by the FCs are numbered as described in Table 1. The state of FC exhibiting the smaller (more negative) and greater (more positive) correlations than the healthy control is termed under- and over-connectivity, respectively. Specific FCs identified for (**a**) ASD [105], (**b**) SSD [116], and (**c**) MDD classifiers [117], are illustrated. These three classifiers are summarized in (**d**), where they did not overlap.

**Figure 3 biomedicines-09-00403-f003:**
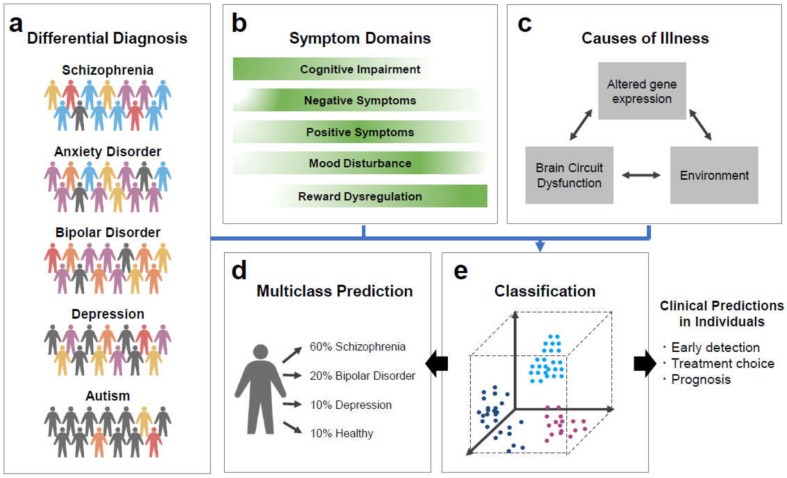
Challenges for precision medicine in psychiatry by artificial intelligence. (**a**) Traditional psychiatric research investigates a given patient group by comparison against the healthy group, possibly creating artificial dichotomies. (**b**) Each psychiatric disorder has various types of the symptoms with their varying degrees. (**c**) The interplay between altered gene expression and neural circuit, and environment such as stress, may elicit psychiatric manifestations. Instead of relying on diagnostic category, a given patient can be classified based on biological and pathological properties (i.e. endophenotypes). (**d**) Artificial intelligence such as machine learning can be extended to compare observations from numerous groups in the same statistical estimation. (**e**) Machine learning algorithms, such as SVM, can automatically extract unknown patterns of variations in individuals simultaneously from heterogenous data labeled with traditional diagnosis. Predictive models could improve patient care by early detection, treatment choice, and prognosis.

**Table 1 biomedicines-09-00403-t001:** The identified terminal regions in ASD/SSD/MDD classifiers. The terminal regions identified for ASD/SSD/MDD classifiers are numbered. See Figure 2.

No	Anatomical Name	No	Anatomical Name
1	anterior lateral fissure	36	posterior sub-central ramus of the lateral fissure
2	anterior ramus of the lateral fissure	37	calloso-marginal posterior fissure
3	diagonal ramus of the lateral fissure	38	collateral fissure
4	anterior sub-central ramus of the lateral fissure	39	intraparietal sulcus
5	calloso-marginal anterior fissure	40	secondary intermediate ramus of the intraparietal sulcus
6	calcarine fissure	41	insula
7	superior postcentral intraparietal superior sulcus	42	paracentral lobule central sulcus
8	primary intermediate ramus of the intraparietal sulcus	43	central sylvian sulcus
9	parieto-occipital fissure	44	cuneal sulcus
10	lobe occipital	45	anterior interior frontal sulcus
11	central sulcus	46	intermediate frontal sulcus
12	subcallosal sulcus	47	median frontal sulcus
13	inferior frontal sulcus	48	polar frontal sulcus
14	internal frontal sulcus	49	sulcus of the supra-marginal gyrus
15	marginal frontal sulcus	50	posterior intra-lingual sulcus
16	orbital frontal sulcus	51	internal occipito-temporal lateral sulcus
17	superior frontal sulcus	52	posterior occipito-temporal lateral sulcus
18	anterior intralingual sulcus	53	olfactory sulcus
19	anterior occipito-temporal lateral sulcus	54	internal parietal sulcus
20	median occipito-temporal lateral sulcus	55	transverse precentral sulcus
21	occipito-polar sulcus	56	intermediate precentral sulcus
22	orbital sulcus	57	median precentral sulcus
23	superior parietal sulcus	58	superior postcentral sulcus
24	interior precentral sulcus	59	rhinal sulcus
25	marginal precentral sulcus	60	posterior inferior temporal sulcus
26	superior precentral sulcus	61	superior temporal sulcus
27	inferior rostral sulcus	62	superior terminal ascending branch of the superior temporal sulcus
28	anterior inferior temporal sulcus	63	sub-parietal sulcus
29	polar temporal sulcus	64	Thalamus
30	anterior terminal ascending branch of the superior temporal sulcus	65	Amygdala
31	paracentral sulcus	66	Accumbens
32	ventricle	67	Caudate
33	posterior lateral fissure	68	Pallidum
34	ascending ramus of the lateral fissure	69	Putamen
35	retro central transverse ramus of the lateral fissure	70	Vermis

## Data Availability

Data sharing not applicable. No new data were created or analyzed in this study.

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
