# Peer review of "Psychiatric Neural Networks and Precision Therapeutics by Machine Learning"

_biomedicines, 2021, doi:10.3390/biomedicines9040403_

Round 1

Reviewer 1 Report

In this review paper it appears clearly that the Authors made a great effort to overview a broad number of psychiatric disorders and the contribution of machine learning approaches to this field. This combination is certainly interesting for the readership of the journal. The paper reads well and the Authors covered a very large amount of significant literature citations.

I have only a minor note: the disorders mentioned in the manuscript could be referred to the DSM-5 and/or DSM-4 (see https://pubmed.ncbi.nlm.nih.gov/20636112/).  Only on [Page 6 Line 4] DSM-5 is clearly mentioned.  DSM-4 is still used by many practitioners, but autistic spectrum disorders and ADHD, for example, tend to be analyzed, in the most recent reports, following dimensional critera set by DSM-5 (see, e.g. https://pubmed.ncbi.nlm.nih.gov/31423596/ and https://pubmed.ncbi.nlm.nih.gov/31768375/)

I suggest to accept the paper with minor revisions.

MINOR REVISION

Discussion: please extend your discussion taking into account the weaknesses/strengths of the new trend of dimensional psychiatry (introduced by DSM-5) and discuss the consequences (advantages and disadvantages) for a machine learning approach.

Author Response

I really appreciate your opinion. Thus, I briefly added its comment in conclusion section in our manuscript. In your context, we already include and discuss incorporation of human imaging results in the main body of this manuscript that can reinforce weakness of RDoC and DSM-5.

Reviewer 2 Report

The paper aims to introduce machine learning methods to clinicians, specially for Psychiatric analysis. Learning methods are the basis for artificial inteligence applications and they have introduced important progress in neuroscience research, however, the manuscript lack in originalty and does not introduce new insights in the area.

Just typing in academic Google, we can find interesting papers (not included in the paper):

[2018] Bzdok et.al., "Machine Learning for Precision Psychiatry: Opportunities and Challenges"
[2019] Durstewitz et.al., "Deep neural networks in psychiatry"
[2020] Zhou et.al., "Machine learning methods in psychiatry: a brief introduction"

I recommed a major revision, updating the references and adding more clinical results in psychiatric using machine learning.

Author Response

The papers you find here have introductive, basic, and ordinal contents suited well for beginners. That is why we have no reason to quote them.

As described in our manuscript, the advanced approaches with a unique combination of machine learning algorithms across multiple sites are needed to avoid the over-fitting and the effects of nuisance variables (NVs). Such types of clinical results are still very few and rare. Thus, we here focus on those clinical studies and results that are good enough to reflect and discuss the current clinical status using machine learning. That is why we have no reason to add more clinical results.

This manuscript is a resubmission of an earlier submission. The following is a list of the peer review reports and author responses from that submission.

Round 1

Reviewer 1 Report

Although the review accounts for a large body of research, I find it largely unfocused and lacking coherence. It is not straightforward to appreciate a combination of different threads that only have psychiatric diseases in common. For example, the authors comprehensively argue for deficiency in decision making as a key biomarker and at the same time discuss the use of machine learning for precision therapeutics or alterations (detection of those too) of brain's network connectivity in brain diseases. It is hard to see a thematic backbone. 

An example of this problematic incoherency is how the introduction is structured. It is confusing for the reader to figure out what the authors review in their manuscript. It is just a combination or mix of loosely connected threads. 

Finally, as for the use of machine learning to study brain connectivity for diagnostic purposes or in precision therapeutics, the authors could more comprehensively cover this broad theme.  Overall, a clear scope, focus and consistency in the reviewed theme is strongly recommended.

Reviewer 2 Report

Here, the authors provide an overview of several DSM characterized disorders and review attempts to leverage machine learning approaches to better characterize these disorders. While this is certainly a worthy topic of review, the large majority of text does not relate to machine learning approaches. There are many spelling errors, grammatical misuse, and some factual misrepresentations in the text that detract from the authors framework. Examples: algorisms (algorithms), individual’s (individuals'), Reinforcement learning investigates how actions in one’s environment (such as treatment) change behaviors (respectfully, this is not the correct definition of reinforcement learning algorithm). In addition, it is problematic that ASD is grouped with psychiatric conditions without much discussion of the significant differences between these illnesses.

I would recommend the authors refocus their work on the most relevant sections, much of the existing text would be better summarized by directing readers to existing high quality works.

Reviewer 3 Report

This paper presents a very important review on a hot topic such as the application of neural networks and machine learning to the study of neural and psychiatric disorders. The authors cover a broad range of areas and disorders, with a particular focus on the potential prediction of therapeutic outcome for pharmacological treatments.

The authors present some specific applications of their approach and highlight the important of decision-making studies. The paper is well written and the only criticism could be limited to the fact that there are several studies aimed towards similar goals that have been overlooked in the past, but very strongly related to the matter of this review. The authors should conside also to include the following papers:

[1] Reciprocal projections in hierarchically organized evolvable neural circuits affect EEG-like signals, Brain Res. 2012, 1434:266-76. doi: 10.1016/j.brainres.2011.08.018.

[2] Predicting neuronal dynamics with a delayed gain control model. PLoS Comput Biol. 2019, 15:e1007484. doi: 10.1371/journal.pcbi.1007484.

[3] A computational perspective on autism, Proc Natl Acad Sci USA 2015, 112:9158-65. doi: 10.1073/pnas.1510583112.